# GP88/PGRN Serum Levels Are Associated with Prognosis for Oral Squamous Cell Carcinoma Patients

**DOI:** 10.3390/biology10050400

**Published:** 2021-05-04

**Authors:** Thomas Greither, Tina Steiner, Matthias Bache, Ginette Serrero, Sven Otto, Helge Taubert, Alexander W. Eckert, Matthias Kappler

**Affiliations:** 1Center for Reproductive Medicine and Andrology, Martin Luther University Halle-Wittenberg, 06120 Halle (Saale), Germany; steiner.tina@web.de; 2Department of Radiotherapy, Martin Luther University Halle-Wittenberg, 06120 Halle (Saale), Germany; matthias.bache@medizin.uni-halle.de; 3A&G Pharmaceutical Inc., Columbia, Maryland, MD 21045, USA; gserrero@agpharma.com; 4Program in Oncology, University of Maryland Greenebaum Comprehensive Cancer Center, Baltimore, MD 21201, USA; 5Department of Oral and Maxillofacial Plastic Surgery, Martin Luther University Halle-Wittenberg, 06120 Halle (Saale), Germany; sven.otto@uk-halle.de (S.O.); matthias.kappler@uk-halle.de (M.K.); 6Clinic of Urology and Pediatric Urology, FA University Hospital Erlangen-Nürnberg, 91054 Erlangen, Germany; helge.taubert@uk-erlangen.de; 7Department of Oral and Maxillofacial Plastic Surgery, Klinikum Nürnberg Breslauer, Paracelsus Medical University, 90471 Nürnberg, Germany; Alexander.Eckert@klinikum-nuernberg.de

**Keywords:** GP88, progranulin, serum, OSCC, prognosis

## Abstract

**Simple Summary:**

An oral squamous cell carcinoma (OSCC) is a tumor of the oral cavity that has a five-year survival rate of only around 50%. As this rate has not increased in recent decades, despite improvements in diagnosis and therapy, novel, easily accessible biomarkers for prognosis assessment are still needed. In our study, we measured the growth factor protein progranulin/GP88 in the serum of OSCC patients and demonstrated that an increased serum GP88 level is associated with a better prognosis for the OSCC patients in our study group. Furthermore, serum GP88 levels were not significantly associated with age, sex, or the tumor’s histological features, indicating that serum GP88 levels may be an independent predictor of an individual OSCC patient’s prognosis. These findings may help to improve therapy management of an OSCC in personalized medicine.

**Abstract:**

Progranulin (PGRN)/GP88 is a growth factor that is expressed in a wide range of tumor tissues. The secreted form is involved in various biological processes including proliferation and inflammation. In several tumor types, the serum GP88 level is associated with a patient’s prognosis; however, data for oral squamous cell carcinomas (OSCCs) have not yet been reported. We measured the serum GP88 levels in 96 OSCC patients by an enzyme immunosorbent assay (EIA) and correlated these data with clinicopathological parameters and patient outcomes. The GP88 levels in the serum of OSCC patients and healthy volunteers were comparable. In OSCC patients, the levels did not correlate with age, sex, or TNM status. In a Kaplan–Meier survival analysis, a serum GP88 level < 68 ng/mL was significantly associated with worsened survival (*p* = 0.0005, log-rank-test) as well as in uni- and multivariate Cox regression analyses (RR = 4.6 [1.6–12.9], *p* = 0.004 and RR = 4.2 [1.2–12.0], *p* = 0.008). This effect was predominant in OSCC patients older than 60.5 years (*p* = 0.027), while in younger patients no significant association between serum GP88 levels and prognosis could be observed. Altogether, lower serum GP88 levels are significantly associated with a worsened outcome for an OSCC and may be an interesting candidate for risk stratification during OSCC therapy.

## 1. Introduction

An oral squamous cell carcinoma (OSCC), which mainly comprises cancer of the oral cavity and adjacent histological structures, is the fifteenth most-diagnosed cancer worldwide [1]. The standard therapy involves surgical excision of the tumor following radiotherapy or chemotherapy. For follow-up rehabilitation of functional defects. the grafting of bone material from different sources [2,3] and patient-specific implantation of prosthetics is practiced [4]. The application of stem cells to reverse bone degeneration or dental loss offers great hope for the future [5,6]. Nonetheless, the five-year survival rate of OSCC patients is only about 50% and has stagnated in recent decades [7]. Therefore, very early detection and prognosis stratification may be the main step for personalized risk management and subsequent therapy management for OSCC patients. 

Liquid biopsies offer the advantage of low-invasive detection of a wide variety of metabolic products or molecular factors generated by the tumor. Given the applicability of glycoprotein 88 (GP88) to serum measurements in a wide variety of body fluids [8], soluble GP88/progranulin (PGRN) may be an interesting candidate for a biomarker in OSCC patients. Furthermore, because links already exist between serum GP88 levels and disease progression or prognosis in malignancies (e.g., breast cancer [9,10,11], non-small-cell lung cancer [12] and prostate cancer [13,14]) and in the genesis and progression of other diseases (e.g., frontotemporal lobar degeneration [15,16], obesity [17,18], systemic lupus erythematosus [19,20] and rheumatoid arthritis [21,22]), changes in serum GP88 levels seem to reflect a pathophysiological state in a wide variety of conditions.

Progranulin/GP88 is a member of the unique growth-factor family, expressed as a soluble 88 kDa glycoprotein and identified with others in the teratoma PC cell line [23]. It contributes to disease progression by stimulating proliferation and invasion in a multitude of different tumor entities [24,25,26,27,28] via the activation of several tumor-related pathways like ERK and TNFR2/Akt signaling [29,30,31,32] and PI3K [32] or Wnt [33] signaling. GP88 can promote angiogenesis via stimulation of VEGF in breast cancer and colorectal cancer cells [29,34]. An accumulation of GP88 in cancer stem cells of hepatocellular carcinoma or an association with cancer stemness in glioblastoma has been suggested [35,36]. GP88 expression has also been reported in relation to different cancer treatment resistance mechanisms such as letrozole and tamoxifen resistance in breast cancer cells [27,34], temozolomide resistance of glioblastoma [35], dexamethasone resistance in multiple melanoma [37], and chemoresistance to cisplatin in ovarian cancer cell lines [38]. Furthermore, GP88 is involved in inflammatory processes by, for instance, suppressing TNFα-induced IL-8 production in human epithelial cell lines [39] or human aortic smooth muscle cells [30]. However, GP88 action on inflammation is a double-edged sword, as the elastin-mediated processing of progranulin to lower molecular weight granulins results in pro-inflammatory activity of the cleaved factor by inducing, for instance, interleukin 8 (IL-8) secretion in epithelial cell lines, which attracts neutrophils [39]. Additionally, the macrophages of GP88-deficient mice were demonstrated to produce increased pro-inflammatory cytokines like monocyte chemoattractant protein-1 (MCP1), IL-6, IL-12 and tumor necrosis factor (TNF) in reaction to an acute infection, while the anti-inflammatory IL-10 production was largely suppressed [40]. Taken together, because GP88 acts at the crossing of several biological processes associated with the pathophysiology of malignancies, it is an interesting target for diagnostic and therapeutic applications.

The aim of this study was to measure GP88 levels in the serum of 96 OSCC patients and correlate these data with the demographic, clinicopathological and prognostic data of the patients to determine whether serum GP88 levels are associated with OSCC occurrence and progression.

## 2. Materials and Methods

### 2.1. Patients

Ninety-six patients suffering from an oral squamous cell carcinoma (OSCC) were included after a positive evaluation of the study by the local ethics committee of the Medical Faculty of the Martin Luther University Halle–Wittenberg (Ethical registry 210/19.08.09/10; approval date: 19.08.2009). The patients gave written informed consent. The median age of the patients was 60.5 years (range: 42–87), with 69 (71.9%) men and 27 (28.1%) women enrolled. Blood sampling was performed in all cases in this study on the day of the surgical excision of the tumor with no prior treatment. Twenty-six patients (27.1%) received radiotherapy subsequent to excision, 11 patients (11.5%) received chemotherapy with cisplatin, and one patient received taxol (1.0%). The patient group is part of the previously described cohort [41,42]. Demographic and clinicopathological data of the patients are summarized in Table 1. Healthy volunteers were blood donors with a median age of 64.0 years (23 males and 10 females).

### 2.2. Preanalytics

Ten milliliters of venous blood was drawn on the day of surgical excision of a patient’s tumor. Blood was immediately centrifuged at 400× *g*, and serum was then transferred to separate reaction tubes and subsequently stored at −80 °C. A serum aliquot was thawed directly before enzyme immunosorbent assay (EIA) measurement.

### 2.3. GP88 EIA

Serum GP88 levels were measured by a quantitative GP88 sandwich EIA developed and manufactured by A&G Pharmaceutical Inc. (Columbia, MD, USA) according to the manufacturer’s protocol (also described in [8]). Briefly, 96-well EIA plates (Corning, Kennebunk, ME, USA) were coated with the antihuman GP88 6B3 monoclonal antibody the day before incubation with 200 μL patients serum (1:10) each for 2 h. After washing, the wells were incubated with rabbit polyclonal 37 kDa antibody as a detection antibody for 1 h. EIA reaction was triggered by TMB substrate incubation for 5 to 15 min and measured by an absorbance readout at 620 nm on a GENios Microplate Reader (Tecan, Männedorf, Switzerland). Quantification was performed by a comparison of the absorption readouts with the control samples (human GP88 at levels from 0 to 20 ng/mL). Standard samples, patients’ samples, and healthy volunteers’ samples were measured in duplicate, and the mean value was applied for concentration calculations.

### 2.4. Statistics

Statistical analyses were performed using SPSS Statistics 25 (IBM, Ehningen, Germany). ROC curves were performed for the comparison of serum GP88 levels of healthy volunteers against OSCC patients and for the assessment of an optimized cut-off value through Youden index calculation. The distribution of serum GP88 levels in different groups (age, sex, T, N, G) were compared by a ChiSquare test and non-parametric tests (Mann–Whitney U-Test, Kruskal–Wallis test, Shapiro–Wilk test). Survival analyses were calculated by Kaplan–Meier analyses and uni-/multivariate Cox regression analyses. Overall survival (OS) was assessed from the date of serum collection to the last follow-up date or death of the patient.

## 3. Results

### 3.1. GP88 Levels Are Not Different in the Serum of OSCC Patients and Healthy Volunteers

We compared the levels of serum GP88 in OSCC patients to those assessed in healthy volunteers. The median age in both groups was not significantly different (OSCC patients: 60.5 years; healthy: 64.0 years; *p* = 0.10; unpaired Student’s t-test). Median serum GP88 levels did not significantly differ in healthy blood donors in comparison to age-matched OSCC patients (57.6 ng/mL vs. 52.1 ng/mL; *p* = 0.61, Student’s t-test; see Figure 1).

### 3.2. Serum GP88 Level and Demographic and Clinicopathological Parameters

We analyzed the GP88 serum levels of the OSCC patients according to sex, age, T and N stage. The levels were slightly lower in female OSCC patients compared those of males (median GP88 level 44.7 ng/mL vs. 55.6 ng/mL); however, this difference was not significant (*p* = 0.249; Mann–Whitney U-Test; see Figure 2a). Serum GP88 levels were also gradually lower in patients ≥60.5 years (median GP88 level 48.1 ng/mL vs. 58.5 ng/mL in younger patients); however, this difference also did not reach significance (*p* = 0.408; Mann–Whitney U-Test; see Figure 2b). Regarding the T stage of the tumor, median serum GP88 levels were almost equal in low-grade tumors (T1/2) compared with advanced tumors (T3/T4; 51.2 ng/mL vs. 52.9 ng/mL; *p* = 0.847; Mann–Whitney U-Test; see Figure 2c). Patients with no nodal metastases exhibited a gradually higher serum GP88 level (55.6 ng/mL vs. 49.9 ng/mL); however, this difference was also not significant (*p* = 0.421; Mann–Whitney U-Test; see Figure 2d). In summary, serum GP88 levels in OSCC patients did not seem to be correlated with age, sex, T or N stage.

### 3.3. Lower Serum GP88 Levels Are Significantly Associated with Worsened Survival

For subsequent survival analyses, an optimized cut-off according to the Youden index calculated from a ROC analysis (see Figure 3), with survival of the patient as the discriminative event, was applied.

The cut-off value exhibiting the highest sensitivity and specificity in the ROC analysis was 68.0 ng/mL, with 65 patients (67.7%) exhibiting a lower serum GP88 level and 31 patients (32.3%) having a higher serum GP88 level. In the Kaplan–Meier survival analyses, a serum GP88 level below 68 ng/mL was significantly associated with shortened mean survival (75.8 months (95% CI: 60.2–91.4 months) vs. 84.8 months (95% CI: 74.6–95.1 months); *p* = 0.002, log-rank test (see Figure 4a).

Furthermore, in an univariate Cox regression analysis, a lower serum GP88 level in the OSCC patients was associated with a 4.6-fold increased risk of death (95% CI: 1.6–12.9; *p* = 0.004; see Figure 4b). Concordantly, in multivariate Cox regression analyses (adjusted for tumor grade (G), tumor stage (T) and lymph node stage (N) of the tumor), lower serum GP88 levels were significantly associated with a 4.2-fold increased risk of death in the patient cohort (95% CI: 1.4–12.0; *p* = 0.008; see Figure 5; see also Appendix A and Appendix A). In TCGA analyses regarding the prognostic impact of the OSCC PGRN mRNA expression on the patients’ outcome, no association could be detected (see Appendix A). We hypothesized that the PGRN mRNA levels in the tumor tissue were not related to the corresponding PGRN serum levels; however, this matter warrants further studies.

### 3.4. Lower Serum GP88 Levels Are a Negative Prognostic Marker in Elderly OSCC Patients

Finally, we analyzed the association of serum GP88 levels with OSCC patient survival in younger (<60.5 years) and older (≥60.5 years) individuals. Age-adjusted cut-off values for both groups were calculated by Youden index optimization, which gave an unchanged cut-off value for the younger patients (68.0 ng/mL) while the cut-off for older patients was lower (65.0 ng/mL). Uni- and multivariate Cox regression analyses revealed that in the older patient group, the relative risk of death was significantly increased for those with lower serum GP88 levels (RR = 5.74 and 5.45, respectively, see Table 2), while in younger OSCC patients, serum GP88 levels had no significant impact on patient survival.

Additional analysis regarding the prognostic impact of GP88 in females vs males, T1/2 vs. T3/4, N0 vs. N+, and patients with vs. patients without radiotherapy are given in the Appendix A. Interestingly, lower GP88 serum levels were only a significant prognostic factor in males, while in female OSCC patients this connection was not significant, probably due to low total numbers (see Appendix A). In an OSCC exhibiting T1/2 or N0, a lower GP88 serum level was also a significant factor for a worse outcome (see Appendix A), while radiotherapy did not influence the predictive power of serum GP88 levels (see Appendix A).

## 4. Discussion

In this study, we measured the serum GP88 level in patients with an oral squamous cell carcinoma (OSCC) and assessed its impact on their demographic, clinicopathological characteristics and prognosis. In our study, serum GP88 levels were not significantly different in patients with an OSCC (52.1 ng/mL) compared to those of healthy volunteers (57.6 ng/mL). Although limited information is available on GP88 serum levels for patients with an OSCC, compared those of healthy volunteers, this result is in contrast to the available literature on other tumors. Breast cancer patients showed increased GP88 serum levels having mean levels of 40.7 to 45.3 ng/mL compared to 28.7 ng/mL for age-matched healthy volunteers, with no significantly different expression due to age or ethnicity [9]. In non-small cell lung cancer patients, serum GP88 concentrations were higher (mean: 49.9 ng/mL) compared to those of healthy controls (mean: 28.4 ng/mL) [12]. Furthermore, lymphoid malignancies demonstrated higher serum GP88 levels (91.3 ng/mL) than were detected in control subjects (mean: 57.7 ng/mL) [43]. Yamamoto and colleagues also determined the mean serum GP88 level in a cohort of 417 healthy individuals to be 40.1 ng/mL, and observed no significant differences associated with age or sex [21]. However, patients with rheumatoid arthritis (RA) demonstrated higher levels of serum GP88 than were found in healthy volunteers [21]. In prostate carcinoma patients, a mean serum GP88 level of 48.7 ng/mL was determined [13]. In glioma patients, mean PGRN serum levels were higher (74.0 ng/mL) than in healthy controls (29.6 ng/mL) [44]. Concerning oral health, mean PGRN serum levels were also higher in patients with Sjögren’s syndrome in comparison to healthy controls [45]. In line with these findings, PGRN levels accompanied by soluble Oxford 40 ligand (sOX40L) levels were found to be increased in Sjögren’s syndrome patients [46].

We were able to demonstrate a significant association between a lower (< 68 ng/mL) serum GP88 level and worsened survival in OSCC patients. On the other hand, an elevated serum GP88 level at three months after surgery and chemotherapy was associated with worsened survival for ovarian cancer [47]. Therefore, it was suggested that GP88 is an independent prognostic factor for this tumor entity due to a missing correlation with age, stage and tumor grade [47]. Similar results were seen by Carlson and colleagues, who found that a higher serum GP88 level was an independent prognostic factor for ovarian cancer [48]. In hormone receptor-positive breast cancer, serum GP88 levels were positively associated with a risk of recurrence, while in hormone receptor-negative tumors no such correlation was demonstrated [11]. In metastatic breast cancer patients, a serum GP88 level >55 ng/mL was associated with a 4-fold decrease in survival [10]. In chronic lymphatic leukemia, increased serum GP88 levels were significantly correlated with worsened survival [49]. Interestingly, serum GP88 levels remained stable in non-progressing diseases over time, while it increased in patients with progressive CLL in a 20–50-month follow-up [49]. Furthermore, in a diffuse large B-cell lymphoma, a higher serum GP88 level predicted worsened survival [45]. In acute lymphoblastic leukemia, an elevated serum GP88 level was not associated with overall survival, but showed a correlation with disease-free survival [50]. Regarding non-malignancies, in community-acquired pneumonia a higher serum GP88 level was also associated with an earlier patient’s death [51]. Taken together, in all studied tumors except an OSCC, higher serum GP88 levels were associated with worsened survival. One explanation may be that GP88 is linked to metastasis [34,52] in several tumors, which is frequently associated with a higher aggressiveness of the respective tumor. Taken together, there was only one patient in our cohort who had distant metastases while 41 (38.7%) had nodal metastases. However, lymph node metastasis was not associated with patient survival in our cohort. Although it is a reliable and applicable predictor of prognosis in patients suffering from an OSCC [53,54], it might not predict prognosis accurately in all cases. After demonstrating that GP88 is an independent prognostic marker in our cohort, we propose it as a promising additive molecular factor for identifying aggressive OSCC subtypes including in the early stages. On a biological level, increased GP88 expression may have beneficial anti-inflammatory effects because it targets tumor necrosis factor receptors (TNFRs), thereby diminishing TNFα action [55,56] and resulting in decreased tumor aggressiveness. This is in agreement with reports showing that increased salivary TNFα and other cytokine levels are associated with an OSCC and higher disease stages [57,58]. In addition, GP88 was shown to play an anti-inflammatory role in an acute lung injury model at increasing regulatory T cells (Tregs) and interleukin- (IL-) 10 levels [59]. This finding further supports the general hypothesis that GP88 as a multifaceted immune-regulatory molecule is involved in acute and chronic inflammation [60]. Inflammation is one of the hallmarks of cancer [61]. As stated above, GP88 exerts an anti-inflammatory action by binding to TNFR1, which antagonizes the pro-inflammatory action of TNF-α [55], which affects, regulates and augments TGF-β1-induced EMT [62,63]. EMT is part of another hallmark of cancer: invasion and metastasis [61]. GP88 binding to TNFR1 also antagonizes the pro-inflammatory action of IL-1β [60]. Both TNF-α and IL-1β are also considered to be possible indicators of malignant transformation in oral disorders [64]. Furthermore, Liu et al. recently demonstrated, in a murine macrophage cell line, that progranulin (GP88) can significantly inhibit liposaccharide-induced NFkB/MAPK pathways by inhibiting the phosphorylation of NFκB and its nuclear translocation and the phosphorylation of ERK1 [65]. Both pathways are also characteristic of tumor development; however, the underlying crosstalk between tumor cell adaptions and metabolic or immunological pathways in an OSCC warrants further research. In our opinion, both metabolic rearrangement and immune-escape mechanisms, in combination with growth factors like GP88, may play pivotal roles in early oral carcinogenesis.

Additionally, we showed that a lower serum GP88 level is an indicator of a poor prognosis, especially in elderly (≥60.5 years) OSCC patients, but in younger patients the effect was less pronounced. Previously, we were able to demonstrate that lower serum GP88 levels were more frequently found in younger prostate cancer patients, while elevated levels were found in older patients [13]. However, in prostate carcinoma patients, elevated levels were only associated with worsened survival in younger patients [13]. However, there are still less data on the impact of serum GP88 levels on the genesis and progression of malignancies or systemic diseases in different age groups; therefore, further research is needed.

## 5. Conclusions

We assessed for the first time the serum levels of GP88 in patients with an oral squamous cell carcinoma. These patients did not exhibit significantly higher serum GP88 levels than found in healthy volunteers. While serum GP88 expression was not correlated to age, sex, or clinical T or N stage of the tumor, a decreased serum GP88 level was significantly associated with worsened OSCC patient survival. GP88 is a pleiotropic factor that has effects on tumorigenesis and metastasis and has an anti-inflammatory effect. Since the OSCC cases investigated here were either local or regional, it is possible that in this study the effect of GP88 was mainly the result of its inflammatory action. To sum up, we identified GP88 is an independent prognostic factor in an OSCC and as a promising biomarker in combination with both the TNM system and other prognostic markers. Measurement of serum GP88 levels, therefore, might be an interesting option for OSCC risk stratification and therapy management.

## Figures and Tables

**Figure 1 biology-10-00400-f001:**
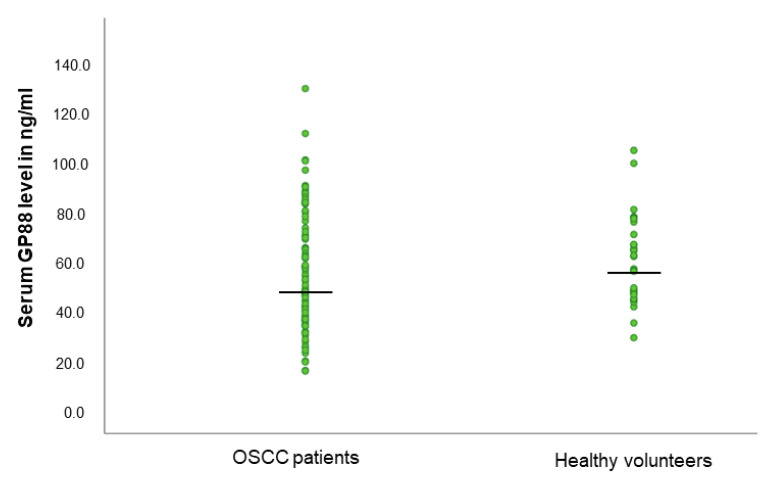
Serum GP88 level distribution in OSCC patients and healthy volunteers.

**Figure 2 biology-10-00400-f002:**
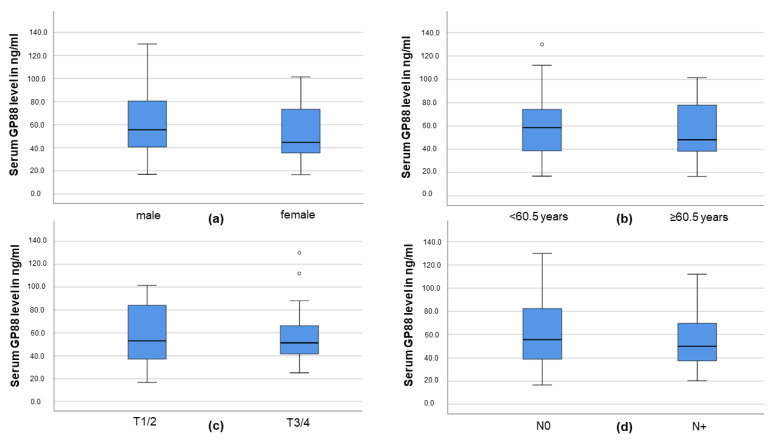
Distribution of serum GP88 levels according to (**a**) sex, (**b**) age median, (**c**) T stage or (**d**) nodal metastasis.

**Figure 3 biology-10-00400-f003:**
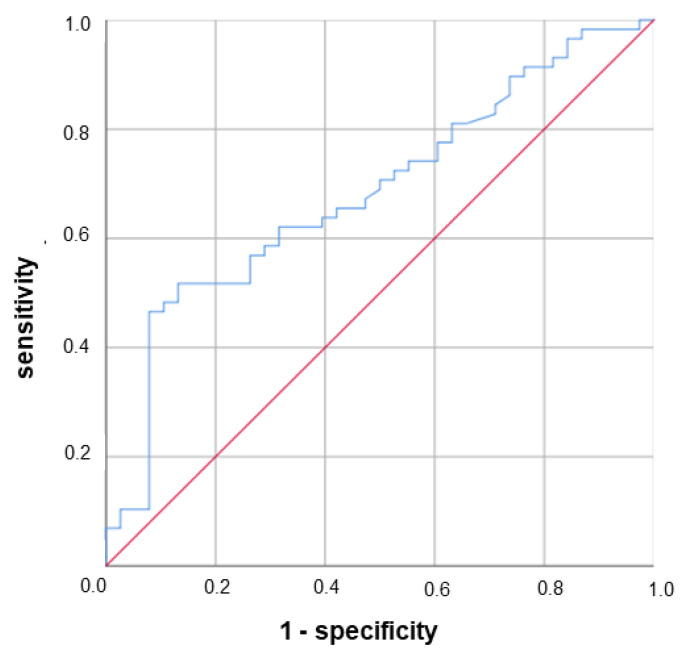
ROC analysis of serum GP88 levels according to overall survival of the OSCC patients used for Youden index calculation. AUC of the ROC curve was 0.684 (95% CI: 0.577–0.792), *p* = 0.002.

**Figure 4 biology-10-00400-f004:**
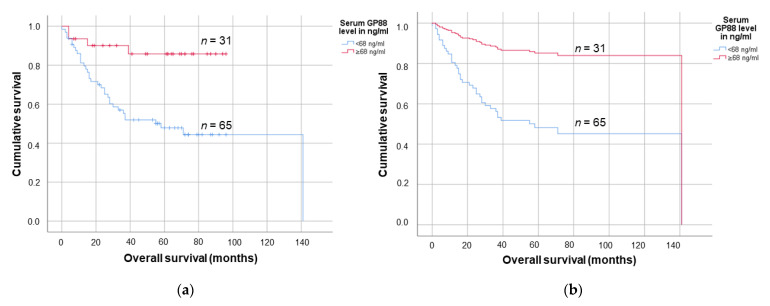
Survival analyses according to (**a**) Kaplan-Meier and (**b**) univariate Cox regression survival analyses estimating the influence of the serum GP88 level on the OSCC patients prognosis.

**Figure 5 biology-10-00400-f005:**
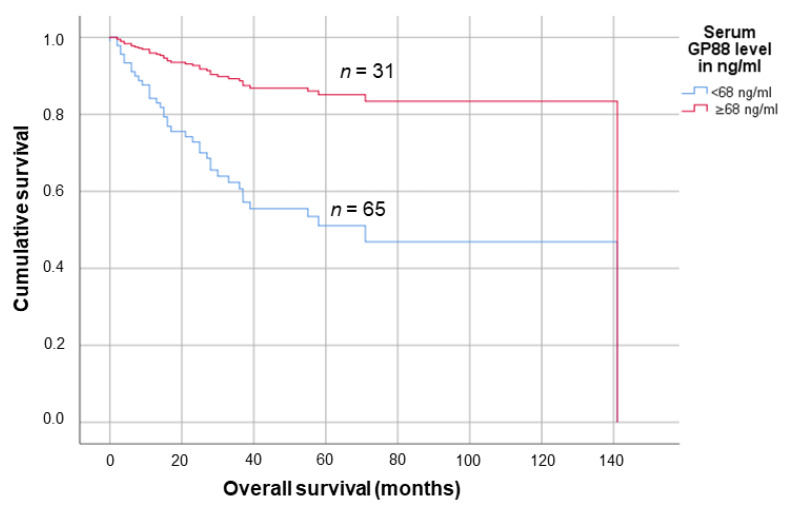
Multivariate Cox regression analysis modelling the survival of OSCC patients according to their serum GP88 levels.

**Table 1 biology-10-00400-t001:** Demographic and clinicopathological data of the OSCC patient cohort. Significant correlation with an increased serum GP88 level is marked in bold.

Parameter	Subgroup	Serum GP88 < 68 ng/mL	Serum GP88 ≥ 68 ng/mL	*n*	*p*
age	<60.5	33	16	49	0.39
	≥60.5	32	15	47	
sex	male	46	23	69	0.73
	female	19	8	27	
patient status	alive	31	27	58	**<0.001**
	deceased	34	4	38	
T stage	T1	12	9	21	0.44
	T2	22	12	34	
	T3	16	4	20	
	T4	15	6	21	
N stage	N0	38	22	60	0.50
	N1	10	2	12	
	N2	16	7	23	
	N3	1	0	1	
M	M0	64	31	95	0.49
	M1	1	0	1	
Grading	1	10	4	14	0.93
	2	40	19	59	
	3	15	8	23	
radiotherapy	no	43	27	70	0.03
	yes	22	4	26	
chemotherapy	no	54	30	84	0.06
	yes	11	1	12	

Significant correlation with an increase serum GP88 level is marked in bold.

**Table 2 biology-10-00400-t002:** Uni- and multivariate Cox Regression analyses on the impact of serum GP88 levels in younger (<60.5 years) and elderly (≥60.5 years).

OSCC Patients	GP88 Levels	Univariate Cox Regression Analysis			Multivariate Cox Regression Analysis		
		*p*	RR (95% CI)	*n*	*p*	RR (95% CI)	*n*
OSCC patients < 60.5 years	Serum GP88 level < 68.0 ng/mL	0.118	3.26 (0.74–14.39)	33	0.261	2.4 (0.52–11.24)	33
	Serum GP88 level ≥ 68.0 ng/mL	reference		16	reference		16
OSCC patients ≥ 60.5 years	Serum GP88 level < 65.0 ng/mL	0.019	5.74 (1.33–24.72)	32	0.027	5.45 (1.22–24.39)	32
	Serum GP88 level ≥ 65.0 ng/mL	reference		15	reference		15

Abbreviations: CI, confidence interval.

## Data Availability

Data can be obtained from the authors on reasonable behalf.

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
