# Peer review of "GP88/PGRN Serum Levels Are Associated with Prognosis for Oral Squamous Cell Carcinoma Patients"

_biology, 2021, doi:10.3390/biology10050400_

Round 1
Reviewer 1 Report
N/A
Author Response
We want to thank the reviewer for the time and effort put in the revision of our manuscript.
Best regards.
Thomas Greither (in behalf of the authors)
Reviewer 2 Report
The manuscript is a revision of previously submitted. The manuscript is only modestly improved from the previous version. Not all the points have been addressed satisfactorily; particularly the comment on “the expression of GP88 in OSCC tissue compared to healthy subjects”. Moreover, due to calculation error in the earlier version it was shown that GP88 level in serum was significantly higher in OSCC subjects compared to healthy. But, in the present version (after recalculation) it was observed that GP88 levels in OSCC subjects are not different than that in healthy subjects, and thus, the significance of (and enthusiasm for) this study is greatly reduced.
The revised data also raised few questions and warrant some additional data to support:
- The levels of GP88 in OSCC and healthy subjects are very similar, yet the author have found that “increased serum GP88 level is associated with a better prognosis for the OSCC patients” – Need explanation and/or some hypothesis.
- “In TCGA analyses regarding the prognostic impact of the PGRN mRNA on patients out-come, no association could be detected, suggesting that mRNA levels in the tissue and serum levels of PGRN are not related”- This need to be established, that’s why it is necessary to measure the tissue and serum level of GP88 from same subjects (that’s why my original comment on the previous version is still very important).
- The authors have demonstrated a significant association between a lower (<68 ng/ml) serum GP88 level and a worsened survival of the OSCC patients. The 68ng/ml value for the “cut-off” needs to be validated in an independent cohort of subjects ( or a two stage study [discovery/validation] needs to be designed. Since this is the only important finding in the manuscript.
- In few other cancer’s (e.g. breast, lung, glioma etc..) serum GP88 levels are significantly higher than healthy subjects, but in OSCC it is not- this needs some discussion.
Minor comments:
Section 3.1: GP88 is elevated in the serum of OSCC patients- This is not TRUE
Author Response
We want to thank the reviewer for the repeated assessment of our manuscript and the helpful remarks. We understand that the recalculation of the GP88 levels in healthy subjects resulting in similar values as found in our OSCC cohort seems less interesting than the finding of an additional diagnostic marker for OSCC. However, as OSCC is much easier to diagnose than other internal tumor entities, in our opinion a diagnostic marker for OSCC is of lesser clinical interest. There is still our finding of an association of lower GP88 serum levels with poor prognosis. Since this prognostic marker can be evaluated by a minimal invasive intervention (liquid biopsy) and a well-established method (sandwich EIA), we think it has some potential for future prognosis studies in OSCC.
The revised data also raised few questions and warrant some additional data to support:
Q1: The levels of GP88 in OSCC and healthy subjects are very similar, yet the author have found that “increased serum GP88 level is associated with a better prognosis for the OSCC patients” – Need explanation and/or some hypothesis.
A1: We agree with the reviewer, that this result is confusing on the first sight. However, as the tumor does not need to be the exclusive or primary source of GP88 expression and serum secretion, we hypothesize that an elevated GP88 serum level may serve as surrogate of an activated immune system, which may help to control tumor growth and aggressiveness. This hypothesis is somewhat supported by the findings in Yamamoto et al., 2017, where the authors demonstrate highly elevated GP88 serum level in patients with lymphoid malignancies in comparison to healthy volunteers. This view is supported by the involvement of elevated PGRN serum levels in autoinflammatory diseases like systemic lupus erythematosus (Tanaka et al., 2012; Qui et al., 2013) or rheumatoid arthritis (Yamamoto et al., 2014; Cerezo et al., 2015), but also in obesity which is frequently associated with chronic inflammation processes (Youn et al., 2009; Todoric et al., 2012). Additionally, in a murine model of systemic lupus erythematosus, Jing and colleagues demonstrated that PGRN deficiency leads to disturbances in T and B cell immune response (Jing et al., 2019).
Q2: “In TCGA analyses regarding the prognostic impact of the PGRN mRNA on patients out-come, no association could be detected, suggesting that mRNA levels in the tissue and serum levels of PGRN are not related”- This need to be established, that’s why it is necessary to measure the tissue and serum level of GP88 from same subjects (that’s why my original comment on the previous version is still very important).
A2: We agree with the reviewer, that the claim we made in the revision of the manuscript overstretches the results of the TCGA analyses. We also acknowledge that the comparison of tumor GP88 protein expression and serum GP88 levels might be interesting; however, we actually do not have protein from the tumor samples in the respective quality and additionally have established the measurement of GP88 protein expression in the paired tumor samples yet. Therefore, we cannot offer robust data on this remark to date.
To include the reviewer’s critic in the manuscript, we changed the sentence accordingly: “In TCGA analyses regarding the prognostic impact of the OSCC PGRN mRNA expression on the patients’ outcome, no association could be detected. We hypothesize that the PGRN mRNA levels in the tumor tissue are not related to the corresponding PGRN serum levels; however, this question warrants further studies.”
Q3: The authors have demonstrated a significant association between a lower (<68 ng/ml) serum GP88 level and a worsened survival of the OSCC patients. The 68ng/ml value for the “cut-off” needs to be validated in an independent cohort of subjects (or a two stage study [discovery/validation] needs to be designed. Since this is the only important finding in the manuscript.
A3: In our study, we chose 68 ng/ml as cut-off value based on the Youden index optimization performed in the present study cohort.
Based on our patients cohort, we have re-evaluated the 68 ng/ml cut-off value by bootstrapping (based on 10,000 bootstrap replicates) receiving a value of 69.85±6.61 ng/ml (95% CI: 57.65 – 71.85). Applying this cut-off value however would not change the present results in the manuscript, therefore, we decided to stay with the initial cut-off value.
Additionally, we validated the dicscriminative properties of the multivariate Cox regression model using the Dxy measure, which, because we regard the complete model, is equivalent to Somers' D.
Here, we applied a 5-fold cross-validation approach and we can state, that Dxy after optimism correction was only slightly reduced from 0.45 to 0.35 after correction. (cph and validate functions were from the rms package for R).
We suggest for future studies to follow this procedure within their own study cohorts, and we are convinced – given that our results are reproduced in other OSCC patients cohorts - that multiple studies will give a clinical relevant cut-off.
Interestingly, Yamamoto et al., 2017 also demonstrated via ROC analyses that a cut-off value of 68.5 ng/ml was the best to distinguish between control patients (n = 34) and patients with lymphoid malignancies (n = 77); however, of note the serum GP88 levels are significantly elevated in lymphoid malignancies.
Q4: In few other cancer’s (e.g. breast, lung, glioma etc..) serum GP88 levels are significantly higher than healthy subjects, but in OSCC it is not- this needs some discussion.
A4: We agree with the reviewer, that this point needs further analysis. However, given the fact that different commercial ELISAs give highly diverting values (for instance serum GP88 median level with A&G ELISA around 40 – 50 ng/ml; with Yonghui Company ELISA around 10 pg/L) and that the numbers of control probands are commonly low (n around 20 – 30), it is hard to establish a “normal” GP88 serum level from the actual data available. The broad-based data are from Yamamoto et al., 2014, where the median GP88 serum level of 417 healthy control subjects is around 40.1 ng/ml. Given the low number of subjects, our normal GP88 level seems to be near the ones measured by Yamamoto et al., 2014 and 2017, and less near to Tkaczuk et al., 2011 and Edelman et al., 2014. However, in the end, further analyses on the mechanisms of serum GP88 are heavily needed.
|
|
malignancy |
Patients (n) |
GP88 median (ng/ml) |
Controls (n) |
GP88 median (ng/ml) |
Distributor ELISA |
|
Tkaczuk et al., 2011 |
Breast cancer |
189 |
39.3 |
18 |
28.44 |
A&G |
|
Edelman et al., 2014 |
NSCLC |
19 |
49.9+14.7 |
20 |
28.4+5.6 |
A&G |
|
Yamamoto et al., 2014 |
RA/OA |
56/31 |
50.2+11.1; 45.4+6.6 |
417 |
40.1+8.7 |
A&G |
|
Yamamoto et al., 2017 |
Lymphoid malignancies |
77 |
91.3 |
34 |
57.7+9.8 |
A&G |
|
Zhang et al., 2014 |
Sjörgren’s syndrome |
26 |
14.57+7.93 pg/L |
26 |
9.8+5.67 pg/L |
Yonghui Company |
|
Qi et al., 2020 |
Sjörgren’s syndrome |
68 |
10.5 pg/L |
50 |
7.5 pg/L |
Yonghui Company |
|
Carlson et al., 2013 |
Ovarian cancer |
150 |
N/A |
50 (benign adnexal mass) |
N/A |
R&D |
Minor comments:
Section 3.1: GP88 is elevated in the serum of OSCC patients- This is not TRUE
We apologize for the mistake and we changed the subsection heading accordingly: “GP88 levels are not different in the serum of OSCC patients and healthy volunteers”
Reviewer 3 Report
The authors show that low serum level of the protein GP88 is associated with poor outcome in a cohort of 96 OSCC patients. The paper is well written and clear. It lacks an understanding of the role of GP88 in OSCC. It is also difficult to see how helpful this biomarker can be for clinician.
This manuscript is a resubmission of an earlier submission. The following is a list of the peer review reports and author responses from that submission.
Round 1
Reviewer 1 Report
In this manuscript, Greither and colleagues have determined the serum levels of GP88 growth factors in 96 OSCC patients and 33 healthy subjects and concluded that serum levels of GP88 could be an interesting candidate for risk stratification during OSCC therapy. While the study has merit, but the study is not comprehensive enough to be published in its present format.
Specific comments/suggestions:
- Role of GP88 in the context of cancer is missing in the introduction.
- Why GP88 is higher in the serum of OSCC patients?- any hypothesis?
- How is the expression of GP88 in OSCC tissue compared to healthy subjects and/or compared to adjacent normal?
- It would also be helpful to analyze GP88 transcript in TCGA database- to see how it varies.
- How the sample size was determined?
- What was the power of the study?
- In a study by Tkaczuk et al (PMID: 21792312) mean serum levels of GP88 in healthy subjects is 7 +/- 5.8 ng/ml while in the present study the value is 17ng/ml – any explanation?
- Figure 1A: Please express the data as dot plot to better visualize the levels of GP88 for each subjects.
- Figure 1A: Sensitivity and specificity values are missing. What is 95% CI for AUC?
- How the sensitivity/specificity values for GP88 compare with other serum-based biomarkers for OSCC.
- Youden index was calculated from Figure 3, why not from Figure 1B?
- “serum of female OSCC patients in comparison to their male counterparts “ -What does it mean by male counterparts?
Author Response
Firstly, we want to thank the reviewer for the thorough revision of our work as well as the many valuable comments and additional ideas for further analyses. We have included revisions to the manuscript according to the comments as detailed below. Unfortunately, during the thorough review of our results due to the reviewers’ comments, we have detected a calculation error in the standard curve used for the estimation of the GP88 serum levels of the healthy control probands. The re-calculated values are included in the revision of this manuscript, and fit more accurately to the already published literature.
For transparency reasons, we attached the original data excel sheet as pdf including the raw data as well as the calculations (with the wrong as well as the corrected standard curve, marked in red and green) for the internal control to the reviewers. The standard curves of the ELISA for the OSCC patients were also reviewed, these were calculated correctly. Standard curve equations for all measurements are also included in the excel table attached.
We apologize for the mistake, and for the confusion as well as the additional work it caused. In the context of this revision, we are thankful for the thorough peer-review, as it allowed us to track down this mistake before a potential publication.
Q1: Role of GP88 in the context of cancer is missing in the introduction.
A1: We agree that the role of GP88 in the context of cancer was not described comprehensively enough. Therefore, we have added the following part in the Introduction chapter:
GP88 can promote angiogenesis via stimulation of VEGF in breast cancer and colorectal cancer cells [24; 29]. An accumulation of GP88 in cancer stem cells of hepatocellular car-cinoma or an association with cancer stemness in glioblastoma has been suggested [30,31]. GP88 expression has been also reported in relation to different cancer treatment resistance mechanisms, e.g. letrozole and tamoxifen resistance in breast cancer cells [22, 29], temozolomide resistance of glioblastoma [30], dexamethasone resistance in multiple melanoma [32], and chemoresistance to cisplatin in ovarian cancer cell lines [33].
Q2: Why GP88 is higher in the serum of OSCC patients?- any hypothesis?
A2: Unfortunately, the explanation is simple and given in the introductory statement: we underestimated our mean serum levels due to a calculation error. The corrected values are given in the manuscript and fit in the range given in the relevant literature (see first paragraph of the discussion section).
Q3: How is the expression of GP88 in OSCC tissue compared to healthy subjects and/or compared to adjacent normal?
A3: We agree with the reviewer, that this is an interesting research topic. Actually, we do not have RNA and/or protein specimen of OSCC tissue and adjacent tissue (or normal oral endothelial samples) in the necessary quantity and quality to address this issue in the current study. However, it is not clear to which extent the OSCC tumor contributes to the serum GP88 level. Therefore, we would like to address this question in future experimental series.
Q4: It would also be helpful to analyze GP88 transcript in TCGA database- to see how it varies.
A4: We have implemented survival analyses regarding the prognostic effect of the PGRN transcript levels in the tumor tissue of Head and Neck patients in the supplementary figure S2. Briefly, PGRN mRNA did not show a significant association with Head and Neck (OSCC) patients’ survival. We added the sentence “In TCGA analyses regarding the prognostic impact of the PGRN mRNA on patients out-come, no association could be detected, suggesting that mRNA levels in the tissue and serum levels of PGRN are not related (see Figure S2).” to the manuscript.
Q5: How the sample size was determined?
A5: The study we conducted was a retrospective pilot study. The sample size was determined by our collection of serum samples and availability of follow-up data. As the 5 year survival of OSCC patients is around 50%, we aimed to include around 100 patients in this study, which were consecutively included.
Q6: What was the power of the study?
A6: In a post-hoc power analysis, the power of the study was 0.9879 (with the input variables: sample size = 96; alpha = 0.05; hazard ratio = 4.2; pE = 0.4; pA = 0.66).
Q7: In a study by Tkaczuk et al (PMID: 21792312) mean serum levels of GP88 in healthy subjects is 7 +/- 5.8 ng/ml while in the present study the value is 17ng/ml – any explanation?
A7: Unfortunately, the explanation is simple and given in the introductory statement: We underestimated our mean serum levels for the healthy volunteers due to a calculation error. The corrected values are given in the manuscript and fit in the range given in the relevant literature (see first paragraph of the discussion section).
Q8: Figure 1A: Please express the data as dot plot to better visualize the levels of GP88 for each subjects.
A8: We formatted Figure 1A accordingly in the revised version of the manuscript.
Q9: Figure 1A: Sensitivity and specificity values are missing. What is 95% CI for AUC?
A9: Given the corrected distribution of the GP88 serum levels between OSCC patients and healthy blood donors, we have excluded the former Figure 1B from the manuscript, as there is no significant difference between the GP88 levels of these two groups seen in Figure 1. The re-calculated AUC was 0.56 [95% CI: 0.46 – 0.66].
Q10: How the sensitivity/specificity values for GP88 compare with other serum-based biomarkers for OSCC.
A10: Due to the changes described in the introductory statement, a sensitivity/specificity for the GP88 serum levels are not longer applicable for our OSCC patients. Therefore, a comparison GP88 level with other serum-based biomarkers is not of relevance for our present study.
Q11: Youden index was calculated from Figure 3, why not from Figure 1B?
A11: Figure 1B presented the capability of GP88 to discriminate between OSCC and healthy volunteers. Since the GP88 values for the healthy volunteers increased after our recalculation, they did not anymore discriminate between OSCC patients and healthy volunteers. Therefore, Figure 1B and the statements for a diagnostic impact of GP88 were omitted from the revised manuscript.
Figure 3 shows the capability of GP88 to discriminate in the OSCC patient group between patients how died or those that were still alive (or censored). In Fig. 3 a Youden index was calculated, i.e, a GP88 value with high sensitivity and high specificity for this discrimination. This value was 68ng/ml (<68ng/ml vs. ≥68ng/ml). This value was applied in the Kaplan-Meier analysis and in the Cox’s regression analyses.
Q12: “serum of female OSCC patients in comparison to their male counterparts “ -What does it mean by male counterparts?
A12: We apologize for the misleading wording. The sentence was changed to “GP88 level was slightly lower in the serum of female OSCC patients in comparison to male OSCC patients (median GP88 level 44.7 ng/ml vs. 55.6 ng/ml);”

Reviewer 2 Report
the manuscript entitled with "GP88/PGRN serum levels are associated with prognosis for oral squamous cell carcinoma patients" makes a pretty good attempt to find a new predictor for the OSCC. the manuscript is pretty straightforward and simple.
I have the following specific comments:
1.in the abstract, line 7, the authors mentions that "in OSCC patients, serum GP88 level did not correlate with age...", then in line 11, the authors say" the effect was predominant in in elder OSCC patients, while younger patients exhibited no significant association between serum GP88 levels and prognosis", which is kind of confusing;
2. the size and style of the figure legend is not consistent and way too small to read, eg. for figure 2, the y axis and x axis is very hard to read in the print out manuscript.
Author Response
Firstly, we want to thank the reviewer for the thorough revision of our work as well as the many valuable comments for clarifications of the present data. We have included revisions to the manuscript according to the comments as detailed below. Unfortunately, during the thorough review of our results due to the reviewers’ comments, we have detected a calculation error in the standard curve used for the estimation of the GP88 serum levels of the healthy control probands. The re-calculated values are included in the revision of this manuscript, and fit more accurately to the already published literature.
For transparency reasons, we attached the original data excel sheet as pdf including the raw data as well as the calculations (with the wrong as well as the corrected standard curve, marked in red and green) for the internal control to the reviewers. The standard curves of the ELISA for the OSCC patients were also reviewed, these were calculated correctly. Standard curve equations for all measurements are also included in the excel table attached.
We apologize for the mistake, and for the confusion as well as the additional work it caused. In the context of this revision, we are thankful for the thorough peer-review, as it allowed us to track down this mistake before a potential publication.
Q1: In the abstract, line 7, the authors mentions that "in OSCC patients, serum GP88 level did not correlate with age...", then in line 11, the authors say" the effect was predominant in in elder OSCC patients, while younger patients exhibited no significant association between serum GP88 levels and prognosis", which is kind of confusing;
A1: We agree with the reviewer, that the previous wording was confusing. Therefore, we changed l. 11 accordingly: “This effect was predominant in OSCC patients older than 60.5 years (p = 0.027), while in patients younger than 60.5 years no significant association between serum GP88 levels and prognosis could be observed.”
Q2: The size and style of the figure legend is not consistent and way too small to read, eg. for figure 2, the y axis and x axis is very hard to read in the print out manuscript.
A2: We have reformatted Figure 2 and hope that the readability is enhanced in the current version.

Reviewer 3 Report
Major comments:
Even if they are included in the main manuscript, results from Supplementary Tables S2-S5 must be described.
Are blood donors the appropriate healthy controls? In addition, the level of GP88 in controls is especially low in this study compared to other studies, the authors need to address this issue because the difference between OSCC patients from their study and healthy patients from the literature is far less significant.
The authors need to clearly state in the manuscript when (after diagnosis, after first line of treatment etc.) GP88 serum level testing was performed.
Minor comments:
-The Table 1 is not correctly positioned on the page.
Author Response
Firstly, we want to thank the reviewer for the thorough revision of our work as well as the many valuable comments for clarifications of the present data. We have included revisions to the manuscript according to the comments as detailed below. Unfortunately, during the thorough review of our results due to the reviewers’ comments, we have detected a calculation error in the standard curve used for the estimation of the GP88 serum levels of the healthy control probands. The re-calculated values are included in the revision of this manuscript, and fit more accurately to the already published literature.
For transparency reasons, we attached the original data excel sheet including the raw data as well as the calculations (with the wrong as well as the corrected standard curve, marked in red and green) for the internal control to the reviewers. The standard curves of the ELISA for the OSCC patients were also reviewed, these were calculated correctly. Standard curve equations for all measurements are also included in the excel table attached.
We apologize for the mistake, and for the confusion as well as the additional work it caused. In the context of this revision, we are thankful for the thorough peer-review, as it allowed us to track down this mistake before a potential publication.
Q1: Even if they are included in the main manuscript, results from Supplementary Tables S2-S5 must be described.
A1: We thank the reviewer for the suggestion and included the following statement in the manuscript “Interestingly, only in male lower GP88 serum levels were a significant prognostic factor, while in female OSCC patients this connection was probably due to the low total numbers not significant (see Table S2). In OSCC exhibiting T1/2 or N0, lower GP88 serum level were also a significant factor for a worsened outcome (see Table S3 and S4), while radiotherapy did not influence the predictive power of serum GP88 levels (see Table S5).”
Q2: Are blood donors the appropriate healthy controls? In addition, the level of GP88 in controls is especially low in this study compared to other studies, the authors need to address this issue because the difference between OSCC patients from their study and healthy patients from the literature is far less significant.
A2: The blood donor cohort were chosen as control group, as they are only allowed to donate blood without any known malignancies or chronic conditions, and as they were in the same age group as our OSCC patients. Unfortunately, the explanation for the significantly lower GP88 serum levels in our blood donor cohort is simple and given in the introductory statement: we underestimated our mean serum levels in the healthy controls due to a calculation error. The corrected values are given in the manuscript and fit in the range given in the relevant literature (see first paragraph of the discussion section).
Q3: The authors need to clearly state in the manuscript when (after diagnosis, after first line of treatment etc.) GP88 serum level testing was performed.
A3: We tried to clarify this point by including the sentence “Blood sampling was performed in all cases included in this study on the day of the surgical excision of the tumor, with no prior treatment.” in the first paragraph of the Materials and Methods section.
Minor comments:
Q4: -The Table 1 is not correctly positioned on the page.
A4: In the revised version, we fixed
